# Detection of Recombinant Hare Myxoma Virus in Wild Rabbits (*Oryctolagus cuniculus algirus)*

**DOI:** 10.3390/v12101127

**Published:** 2020-10-05

**Authors:** Fábio A. Abade dos Santos, Carina L. Carvalho, Andreia Pinto, Ranjit Rai, Madalena Monteiro, Paulo Carvalho, Paula Mendonça, Maria C. Peleteiro, Francisco Parra, Margarida D. Duarte

**Affiliations:** 1Instituto Nacional de Investigação Agrária e Veterinária, Av. da República, Quinta do Marquês, 2780-157 Oeiras, Portugal; carina.carvalho@iniav.pt (C.L.C.); madalena.monteiro@iniav.pt (M.M.); paulo.carvalho@iniav.pt (P.C.); paula.mendonca@iniav.pt (P.M.); margarida.duarte@iniav.pt (M.D.D.); 2CIISA, Faculdade de Medicina Veterinária, Universidade de Lisboa, Avenida da Universidade Técnica, 1300-477 Lisboa, Portugal; mcpelet@fmv.ulisboa.pt; 3Instituto Universitario de Biotecnología de Asturias (IUBA), Departamento de Bioquímica y Biología Molecular, Universidad de Oviedo, 33006 Oviedo, Spain; fparra@uniovi.es; 4Paediatric Respiratory Medicine, Primary Ciliary Dyskinesia Centre, Royal Brompton & Harefield NHS Trust, London SW3 6NP, UK; a.pinto@rbht.nhs.uk (A.P.); r.rai@rbht.nhs.uk (R.R.)

**Keywords:** myxomatosis, recombinant myxoma virus, ha-MYXV, European rabbit, *Oryctolagus cuniculus algirus*, species jump, spillover

## Abstract

In late 2018, an epidemic myxomatosis outbreak emerged on the Iberian Peninsula leading to high mortality in Iberian hare populations. A recombinant Myxoma virus (strains MYXV-Tol and ha-MYXV) was rapidly identified, harbouring a 2.8 kbp insertion containing evolved duplicates of M060L, M061L, M064L, and M065L genes from myxoma virus (MYXV) or other Poxviruses. Since 2017, 1616 rabbits and 125 hares were tested by a qPCR directed to M000.5L/R gene, conserved in MYXV and MYXV-Tol/ha-MYXV strains. A subset of the positive samples (20%) from both species was tested for the insert with MYXV being detected in rabbits and the recombinant MYXV in hares. Recently, three wild rabbits were found dead South of mainland Portugal, showing skin oedema and pulmonary lesions that tested positive for the 2.8 kbp insert. Sequencing analysis showed 100% similarity with the insert sequences described in Iberian hares from Spain. Viral particles were observed in the lungs and eyelids of rabbits by electron microscopy, and isolation in RK13 cells attested virus infectivity. Despite that the analysis of complete genomes may predict the recombinant MYXV strains’ ability to infect rabbit, routine analyses showed species segregation for the circulation of MYXV and recombinant MYXV in wild rabbit and in Iberian hares, respectively. This study demonstrates, however, that recombinant MYXV can effectively infect and cause myxomatosis in wild rabbits and domestic rabbits, raising serious concerns for the future of the Iberian wild leporids while emphasises the need for the continuous monitoring of MYXV and recombinant MYXV in both species.

## 1. Introduction

In the Mediterranean ecosystems, wild rabbit is an important prey for more than 40 predatory aerial and terrestrial species, some of which are endangered [1]. It also plays a crucial role as a soil “architect”, contributing to seed dispersal, and landscaping [2]. Besides its ecological role, the wild rabbit is an important game species economically and socially.

Myxoma virus (MYXV) and rabbit haemorrhagic disease virus 2 (RHDV2) are the two major pathogen threats for the European rabbit (*Oryctolagus cuniculus*), and may occasionally be found simultaneously [3]. The etiological agent of myxomatosis is MYXV, a double-stranded DNA *Leporipoxvirus* of the family Poxviridae [4].

Myxomatosis is an endemic disease of South American rabbits and was first described in laboratory rabbits in 1898 in Uruguay [5]. The disease is characterised by the presence of nodules in the skin surrounding the eyes, nose, mouth, ears, and genitalia. Conjunctivitis, accompanied by purulent discharge is frequently found as a signal of disease [6].

Despite these signs being the most commonly found in the classic, nodular or typical form of disease, myxomatosis can also be found as a respiratory form (amyxomatous form), with variable degrees of severity, where cutaneous signals are minor or not observed [7,8,9]. The origin of this amyxomatous virus is still unclear. Viral mutations and reactivation of subclinical infections are two of the hypotheses proposed [10].

Regardless, the two disease presentations, myxomatosis was considered a rabbit disease during many decades, with some scarce reports in European hares [11,12].

Accordingly, during a National Leporid Surveillance Program (Project +Coelho, Dispatch 4757/17, 31th may), that started in mid-2017, 92 hares and 903 rabbits, collected until October 2018, were analysed for MYXV-DNA using a qPCR directed to the diploid gene M000.5L/R, which is conserved in MYXV and recombinant MYXV strains. Until this date, no hare was positive for any MYXV strain.

The emergence of myxomatosis in the Iberian hare in mid 2018, was caused by a recombinant myxoma virus (first designated as MYXV-Tol, and subsequently ha-MYXV considering its modified tropism towards hares), harbouring an insertion of about 2.8 kbp [13,14,15,16]. After this, health surveillance in the Iberian hare within the scope of Project +Coelho (investigating MYXV [17], RHDV2 [18] and LeHV-5 [19]) and in the wild rabbit (investigating MYXV and RHDV2 [18]) was extended to include screening for ha-MYXV as described by Dalton et al. [15].

The detection of a recombinant MYXV in hares, and the apparent segregated circulation of classical MYXV in rabbits and ha-MYXV in hares, initially suggested the adaptation of MYXV to hares in order to efficiently multiply in this species. Given that hare MYXV, originally considered hare specific, is also being detected in rabbits, who succumbed to the disease, a more generalist designation, geographic and species independent, such as rec-MYXV (for recombinant myxoma virus), may be preferable for the future.

Until the cases reported here, in all tested samples, classic MYXV was only found in wild rabbits and recombinant MYXV in Iberian hares. To our knowledge, we are reporting for the first time, the detection of myxomatosis in European rabbit caused by the recombinant MYXV, adding concerns to the already fragile conservation state of the wild rabbit, taking into account its threat of extinction [20].

## 2. Materials and Methods

### 2.1. Sample, Necropsy and Histopathology

Two adult wild rabbits (*Oryctolagus cuniculus algirus*), in good body condition, found dead in June 2020 (Male, 15758PT20) and July 2020 (female, 20545PT20, from here named Female 1) in the same hunting reserve in Moura, district of Beja, and one wild rabbit in good body condition, found dead in August 2020 in Samora Correia, district of Santarém (female, 22660PT20, from here named Female 2) were collected and investigated within the scope of a national surveillance program in action since August 2017.

Necropsy was performed according to routine procedures, and samples were collected for bacteriology (liver, spleen and lung), parasitology (gastrointestinal tract and liver), histopathology (lung, liver, spleen, kidney, eyelid and genitalia) and virology (liver, spleen, lung, kidney, eyelid andgenitalia).

For histopathology, skin and genitalia fragments were fixated in 10% neutral buffered formalin (*w/v*), routinely paraffin embedded, sectioned at 4 µm, and stained with Hematoxylin and Eosin (H&E).

### 2.2. Parasitological and Bacteriological Examination

Parasitological examination of the intestine was carried out resourcing to direct wet mount, sedimentation and filtration techniques. Liver, spleen and lung samples were analysed using standard bacteriological methods. Enterobacteriaceae and non-Enterobacteriaceae were tested using the ID 32E (Biomerieux^^®^^, Lisbon, Portugal) test and the API 20NE kit (Biomerieux^®^) test respectively. The presence of Streptococcus and Staphylococcus was investigates using the ID 32 STREPT (Biomerieux^®^) and the ID 32 STAPH kits (Biomerieux^®^), respectively. The API CORYNE (Biomerieux^®^) kit was used for the identification of Corynebacteria and coryne-like organisms. For Salmonella, peptone water and Rappaport Vassiliadis semi solid culture media were used. Whenever there was a suspicion of Salmonella, the agarose SMID2 and XLD culture media were used. Other culture media for bacterial identification in the samples included the MacConkey agar and the Blood agar culture media.

### 2.3. Virological and Serological Examinations

For nucleic acid extraction, fresh samples of liver and spleen, kidney, lung, eyelid and genitalia were homogenised at 20% (*w/v*) with phosphate buffered saline (PBS) and clarified at 3000 *g* for 5 min. Total DNA and RNA were extracted from 200 μL of the clarified supernatants, using the MagAttract 96 cador Pathogen Kit (Qiagen, Hilden, Germany) in a BioSprint 96 nucleic acid extractor (Qiagen, Hilden, Germany), according to the manufacturer’s protocol.

The rabbits were tested for rabbit haemorrhagic disease virus 2 (RHDV2) and MYXV by real time PCR targeting the M000.5 L/R gene [17,18]. The 2.8 kbp insert was investigated by the system described by Dalton et al. [15] using primers 9A/9B and 9E/9F that flank the insertion, allowing the amplification of a 3.1 or 4.6 kbp region in recombinant MYXV or a 300 bp region (absence of insert, using the oligomers 9E/9F) in MYXV. Amplification reactions were carried out in a Bio-Rad CFX96™ Thermal Cycler (Bio-Rad Laboratories Srl, Redmond, USA), using the One Step RT-PCR kit (Qiagen, Hilden, Germany) for RHDV2, and the HighFidelity PCR Master Mix (Roche Diagnostics GmbH, Mannheim, Germany), for MYXV and recombinant MYXV.

A commercial kit (Civtest^®^ Cuni Mixomatosis—Hipra, Girona, Spain) developed for the detection of rabbit MYXV antibodies was validated for hare sera [21] and used to detect MYXV antibodies, following the manufacturer’s instructions. For Female 1, serosanguinolent thoracic fluid was used instead of serum due to blood coagulation.

### 2.4. Sequencing Analysis

The initial PCR products (≈3100 bp or ≈4600 bp) encompassing the 2.8 kbp insert, were visualised in 2% horizontal electrophoresis agarose gel, purified using the NZYGelpure kit (Nzytech, Lisbon, Portugal), and directly sequenced using the ABI Prism BigDye Terminator v3.1 Cycle sequencing kit (Thermo Fisher Scientific, Waltham, MA, USA) on a 3130 Genetic Analyser (Applied Biosystems, Foster City, CA, USA). Sequencing by primer walking was carried out, with a total of 12 additional oligomers designed for the purpose (Table 1).

The two nucleotide sequences obtained (15758PT20 and 20545PT20) were assembled using the Seqscape Software v2.7 (Applied Biosystems, Foster City, CA, USA), and submitted to GenBank (MT940239 and MT940240).

### 2.5. Isolation

Isolation of MYXV from the rabbits’ tissues (15758PT20, 20545PT20 and 22660PT20) was achieved separately from eyelid, genitalia and lung. Samples were homogenised at 20% (*w/v*) in PBS containing penicillin, streptomycin and amphotericin B (antibiotic-antimycotic), used according to the manufacturer (Gibco, Massachusetts, EUA). Following centrifugation (3000× *g*, 10 min), the supernatant was filtered through a 0.45-μm-pore-size filter (Millipore Express, Darmstad, Germany) and used to inoculate sub confluent (70%) Rabbit Kidney (RK13) cells (ATCC-CCL-37). RK13 cells were grown in Eagle´s medium supplemented with 5% foetal calf serum (Gibco), penicillin, streptomycin and amphotericin B (antibiotic-antimycotic used at 1:100, Gibco) and 50 μg/mL gentamicin (Gibco). Cells were maintained at 37 °C under humidified atmosphere with 5% CO_2_ and observed daily for cytopathic effect (CPE) by phase contrast microscopy. The supernatant and cell pellet of each passage were tested for the presence of MYXV by qPCR [17].

The isolation of the virus allowed verifying its viability in the rabbit tissues, inferred by the cytopathic effect and in-house immunofluorescence protocol using MYXV positive hare serum (*protocol available on request*). The photographs were taken using an Inverted research microscope, Nikon Eclipse Ti (Nikon Instruments Europe, Amsterdam, Netherlands).

### 2.6. Electron Microscopy

The fragments selected (eyelid and lung) for transmission electron microscopy (TEM) were placed in 10% buffered formalin (*w/v*). Samples were then washed and transferred to 0.05M cacodylate buffer containing 2.5% glutaraldehyde, and post-fixed with aqueous 1% osmium tetroxide (EMS) for 1 h, fragments were then stained in block with ready-to-use UA-zero (Agar Scientifics, Essex, United Kingdom), after which they were dehydrated in increasing concentrations of ethanol, infiltrated and embedded in Araldite resin (Agar Scientifics). Polymerisation was performed at 60 °C for 2 days. Ultrathin sections were cut using a Reichert ultracut E ultramicrotome (Leica, Wetzlar, Germany), collected to 1% 200 mesh copper grids (Agar Scientifics), and examined in a Jeol 1400plus transmission electron microscope at an accelerating voltage of 120 kV. Digital images were obtained using an AMT XR16 bottom mount digital camera (AMT©, Woburn, MA, USA). The sections were systematically analysed using AMT© software and several high and low magnifications were acquired.

## 3. Results

### 3.1. Necropsy and Histopathology

The Male wild rabbit had mild swelling of the eyelids (Figure 1), Female 1 mild swelling of the eyelids and vulva (Figure 2) and Female 2 nodular thickening of the right ear and erosive lesions in the muzzle. Histopathology of the lungs showed focal alveolar oedema with hyaline substance deposits in the alveolar septa in the Male (Figure 3) and infiltration of alveolar septa by mononucleated cells and focal necrosis of alveolar septa with deposits of hyaline substance in Female 1. There was bacterial infiltration in the lung parenchyma of the Male rabbit. The eyelid of the Male presented oedema with epidermal detachment (Figure 4). Due to autolysis, the histopathologic analysis of Female 2 was impaired.

### 3.2. Virological, Bacteriological and Parasitological Results

The three animals tested negative for RHDV, RHDV2 and LeHV-5. The Cq values obtained with the qPCR targeting the diploid MYXV gene M000.5L/R in the tissues from both rabbits revealed high viral charges in the liver/spleen (Male = 2.01 × 10^9^ copies/mg; Female 1 = 1.88 × 10^9^ copies/mg; Female 2 = 1.83 × 10^10^), lung (Male = 1.3 × 10^10^ copies/mg; Female = 1.53 × 10^9^ copies/mg; Female 2 = 2.31 × 10^9^), eyelid (Male = 1.13 × 10^10^ copies/mg; Female 1 = 2.41 × 10^10^ copies/mg; Female 2 = 1.25 × 10^11^), genitalia (Male = 7.47 × 10^9^ copies/mg, Female 1 = 1.30 × 10^10^ copies/mg; Female 2 = 3.05 × 10^9^) and kidney (Male = not tested; Female 1 = 2.92 × 10^8^ copies/mg; Female 2 = not tested). Only the 4.6 kbp fragment was obtained with the PCR directed [18] to the 2.8 kbp insertion, indicating the presence of recombinant MYXV in the tissues of the rabbits and the absence of classical MYXV.

Serology using a commercial kit (Civtest^®^ Cuni Mixomatosis-Hipra) according to the manufacturer’s instructions, showed a high antibody titer in the Male rabbit (RI10 of 19.6) and in the Female 2 (RI10 of 9.2), similar to the RI values of hare positive control serum. Despite that the RI value (<1.0) obtained for the Female 1 suggests no seroconversion, considering that blood serum was not available, no robust conclusions can be taken.

*Bordetella bronchiseptica* and *Escherichia coli* were isolated from the lungs of the Male and Female 1, respectively. Faeces from rabbits showed small infestations of *Eimeria spp*. oocysts.

### 3.3. Molecular Characterisation

Around position ≈61,000 nt of the complete MYXV genome (Lausanne strain), ORFs M060R, M061R, M062R, M063R M064R, M065R and M066R are sequentially located and close together, in different frames, with ORFs M065R and M066R overlapping by 100 nt (Figure 5A).

Sequencing analysis of the 4.6 kbp amplicon confirmed the presence of an additional 2.8kbp region within the M009L gene around nucleotide position 12,336 in the reference MYXV strain AF170726. M009L split into ORF M009L-a containing the original 5’ end, and ORF M009L-b corresponding to the original 3’ end.

In silico analysis of the 4600/2800 bp sequence showed the presence of five ORFs with some degree of similarity with genes M060R, M061R, M064R, M065R and M066R of MYXV strains, but with inverted orientation (Figure 5B). 

ORF M066R encodes a 185 aa long protein and is found in MYXV (e.g. AAF14954.1). ORF M066L (the remaining of the complete gene M066R) encodes a putative partial protein of 70 amino acids in the recombinant MYXV from Portugal. Despite being present in MYXV-Tol and ha-MYXV strains from Spain [13,15], this ORF was not annotated previously. M066L is recognised between ORF M065L and ORF M009L-b, sharing 80% similarity with the homologous sequence of ORF M066R in the ha-MYXV. This small ORF overlaps the M009L-b ORF by 21 nucleotides and M065 by 101 nucleotides (Figure 5C).

The two nucleotide sequences obtained from the Male and Female 1 wild rabbits were identical to each other and to the homologous sequences from other MYXV-Tol/ha-MYXV (MK836424 and MK340973). The differences between the truncated ORF M066L and the homologous M066R ORFs from MYXV-Tol and ha-MYXV obtained from hares (MK836424 and MK340973), and classical MYXV obtained from rabbits (MK388144, MK388143, MK388142 and MK388141 (MYXV) are shown in Appendix A. In particular, the percentage of similarity between ORF M066L and ORFs M066R despite its species of origin is around 79%.

The M066L sequences obtained are 100% similar to correspondent sequences of recombinant MYXV strains described before (MK836424 and MK340973) in Spain. About 79% of similarity was observed between the M066L and the M066R sequences from other ha-MYXV and MYXV strains.

The putative M066L protein obtained presented 80% identity with M066R protein of MYXV-Tol and ha-MYXV strains described before (MK340973 and MK836424). The same similarity was also observed between the M066L and the M066R sequences from classic rabbit MYXV strains (Appendix A).

### 3.4. Isolation of the Virus in Cell Culture

The successful isolation of the recombinant MYXV in RK13 cells from a separate eyelid, genitalia and lung samples from the Male rabbit and from the eyelid of the Females rabbits, confirmed its viability and infectiousness, proving that the virus was multiplying in the rabbits’ tissues. Viral isolation was confirmed by cytopathic effect (CPE) at day 5 in RK13 subconfluent infected cells, by indirect immunofluorescence of the cells (*protocol available on request*) and by conventional PCR of the cell supernatant.

The characteristic CPE at late stage of infection was observed from day three after inoculation (Figure 6A,C). By qPCR we demonstrated the progressive decrease of the Cq value in DNA samples extracted from cell culture supernatant aliquots, collected at day 1, 5 and 10 *(results not shown)*. To demonstrate the presence of the viral protein in the RK13 cells, an indirect immunofluorescence test was performed at day 3 (Figure 6B) and day 6 post inoculation, allowing to confirm Myxoma virus foci (Figure 6D).

### 3.5. Transmission Electron Microscopy

Analysis by electron microscopy allowed observing poxvirus-compatible particles in the lung and eyelid. In the eyelid, a higher number of viral particles was observed, especially in epithelial cells (Figure 7). The degree of advanced autolysis of the tissues did not allow a more detailed evaluation of the type of cells mostly infected.

## 4. Discussion

The external signs of myxomatosis in three adult rabbits found dead between June and August 2020 in Alentejo that arrived to the National Reference Laboratory for Animal Diseases (INIAV, I.P.) for investigation corresponded to mild to moderate swelling of the eyelids and genitalia. 

Both male and females tested positive to MYXV-DNA by the M000.5 L/R gene qPCR, and by the 2.8 kbp insert PCR, showing infection by the natural recombinant MYXV. None of the rabbits were co-infected with classical MYXV strains, RHDV2, RHDV or LeHV-5.

Most of the wild rabbits that died of myxomatosis, generally arrived to the laboratory with severe swelling of the eyelids and genitals, often accompanied by ocular purulent discharge and very frequently in a state of thinness or cachexia. The high viral loads found in several tissues, and the good body condition of these three wild rabbits, suggest that a shorter course of disease may have taken place. Although further testing is necessary to support this relation, this may imply a possible higher virulence of the natural recombinant MYXV strain towards wild rabbits. With regards to the MYXV strains, the national surveillance plan of wild leporids in action in Portugal since 2017 allowed for the testing of more than 57 infected wild rabbits [22]. A total of 73% of the rabbits found dead with myxomatosis (infected with classic MYXV strains) presented median/poor corporal condition or even cachexia, reflecting the ability of the animals to survive infected for longer periods [22]. A lower adaptation of the recombinant MYXV strains to rabbits, comparing the MYXV classic strains with which rabbits have evolved for more than 50 years [23], may eventually account for these apparent differences.

The detection of a recombinant MYXV circulating in hares, and its apparent segregation from MYXV circulating in rabbits, initially suggested the adaptation of MYXV-Tol and ha-MYXV to hares in order to efficiently multiply in this species.

Sequencing of the 2.8 kbp insert from the two rabbits showed that both recombinant MYXV strains have the same poxvirus gene “cassette” previously described in Iberian hares [3,13,15]. 

However, we described a putative truncated gene similar to the M066R gene of the myxoma virus that is also present, though not annotated, in the Myxoma virus sequences obtained previously from Iberian hares. As in the MYXV-Tol (MK836424) and ha-MYXV genomes (MK340973), M062R and the M063R are not found in the insert.

The origin of this insert was discussed previously by other authors, and is not a goal of this work. However, the putative protein encoded by ORF M066L is 65.22% to 76.81% similar to homologous ORFs in capripoxviruses, cervidpoxviruses, suipoxviruses, yatapoxviruses but not with the BeAn 58058 virus, appointed previously [13] as a potential donor, or sharing an ancestral donor, of the genetic material found in the insert. On the other hand, the higher similarity of putative protein encoded by M066L with rabbit fibroma virus and with classical Myxoma virus strains, suggests that the insert may have originated from one of these viruses, or a similar virus, not yet described.

During the three months in which the three rabbits were collected, only a small number of found dead rabbits with myxomatosis arrived at the laboratory from mainland Portugal, limiting any inference about the prevalence, frequency and distribution of the recombinant MYXV in the wild. However, since the natural recombinant MYXV emergence in 2018, and according to data collected under the +Coelho project, Beja was the district from which more hares were sampled (52 out of 170) and tested, and was also one of the districts most affected by myxomatosis (34.6% of positivity in the sample).

The detection of the recombinant MYXV in wild rabbits raises serious concerns at different levels, constituting an additional treat to the already fragile wild rabbit, which entered to the IUCN’s endangered conservation status last year [20]. If the recombinant MYXV and classical MYXV strains behave as different viruses in rabbit, with no full cross protection between the two, the jump of a recombinant MYXV into the rabbit populations will eventually accelerate the decline of these already diminished wild populations. On the other hand, the fact that the recombinant MYXV affects both the Iberian hare and the wild rabbit, may favour the maintenance of the virus as more hosts are available for virus replication and circulation. The recombinant MYXV may therefore become endemic in the same way that classic strains did, allowing the co-evolution in both species. However, the ability to infect the wild rabbit, may lead the recombinant MYXV to prefer the rabbit host, taking into account the greater dispersion and higher density compared to the Iberian hare, which would facilitate their environment maintenance. Further concerns include the rabbit industry, and the need to evaluate if MYXV or Shope Fibroma virus attenuated vaccines are protective against the recombinant MYXV.

Although vaccination is highly effective in the industry, inducing generally the seroconversion of almost 100% of the animals [24], parenteral vaccination of wild populations is almost impossible. Another major concern arises from the emergence and circulation of this new strain in wild rabbit populations, in which virus-host co-evolution regarding classical MYXV strains occurred over the years [25]. The emergence of new MYXV strains theoretically poses a great risk to the rabbit threatened of extinction.

Since the complete genome sequences were not obtained in this study, there are no certainties that the recombinant MYXV strains found in the three rabbits are identical to MYXV-Tol or ha-MYXV. Therefore, we cannot exclude the existence of other mutations that may have contributed to the ability of the recombinant MYXV to cause disease in rabbits.

## 5. Conclusions

Almost two years after the emergence of a recombinant MYXV in Iberian hares, our findings bring one new piece into the model of host-myxoma virus co-evolution by demonstrating the pathogenicity of this recombinant virus towards rabbits. It is important to continue monitoring the disease in wild rabbits and hares in order to ascertain the geographic dimension of the spillover phenomena or the spread of this jump of recombinant hare MYXV back to the European rabbit.

## Figures and Tables

**Figure 1 viruses-12-01127-f001:**
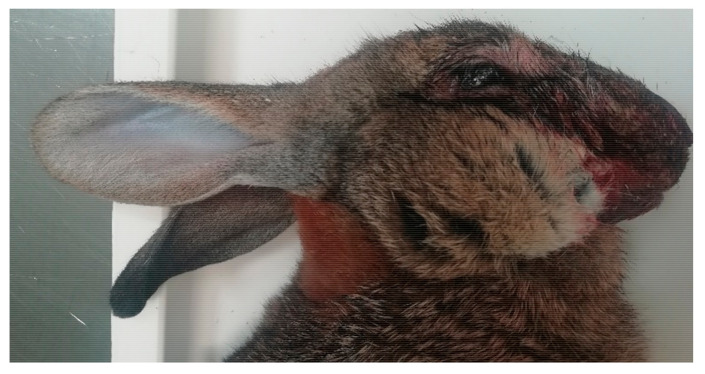
Mild oedema of the eyelid and presence of serous discharge (Male).

**Figure 2 viruses-12-01127-f002:**
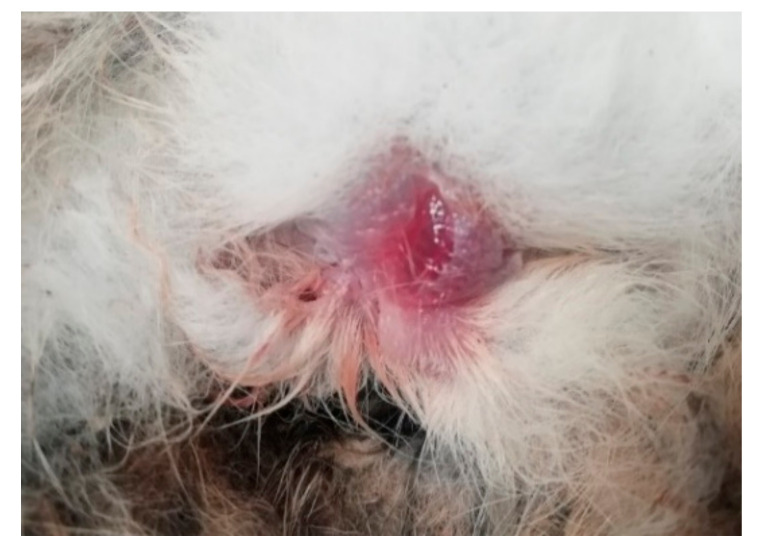
Oedema of the vulva (Female 1).

**Figure 3 viruses-12-01127-f003:**
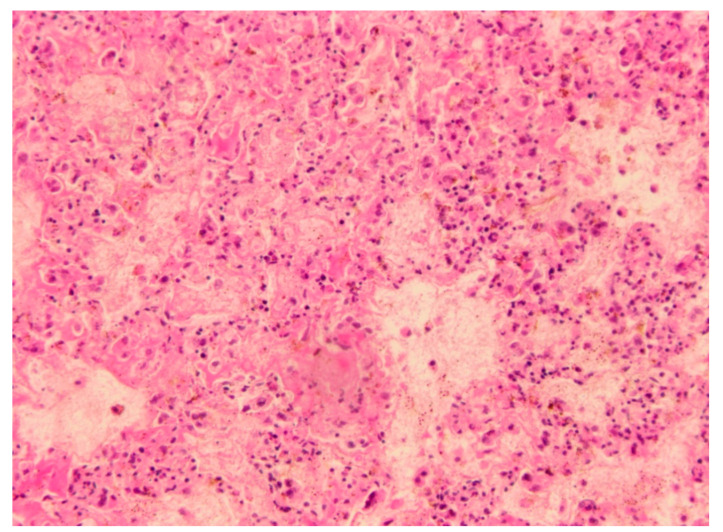
Microscopic finding in Female 1. Infiltration of alveolar septa by mononucleated cells and focal necrosis of alveolar septa with deposits of hyaline substance (H&E, 100×).

**Figure 4 viruses-12-01127-f004:**
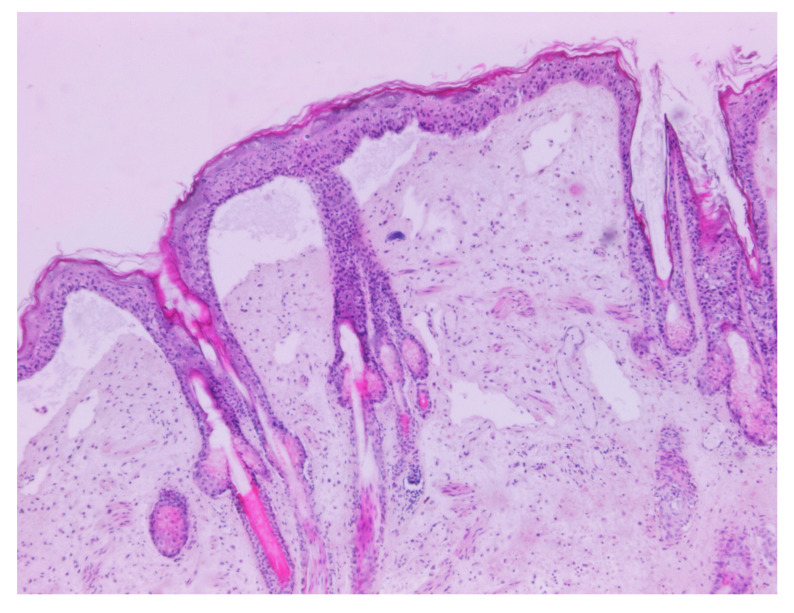
Microscopic finding in Male. Eyelid presenting oedema with epidermal detachment (H&E, 40×).

**Figure 5 viruses-12-01127-f005:**
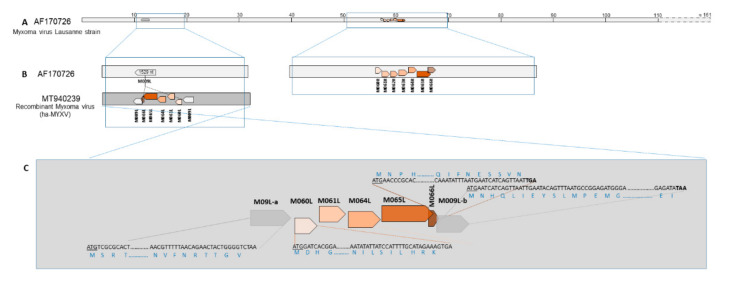
(**A**) Linear genomic organisation of the reference Lausanne strain with the location of ORFs M009L and ORFs M061R and M66R. (**B**) Schematic representation comparing the uninterrupted receptor ORF M009L in the Lausanne strain with the insert in ha-MYXV. (**C**) Detail on the flaking regions of the insert and relative position of the ORFs.

**Figure 6 viruses-12-01127-f006:**
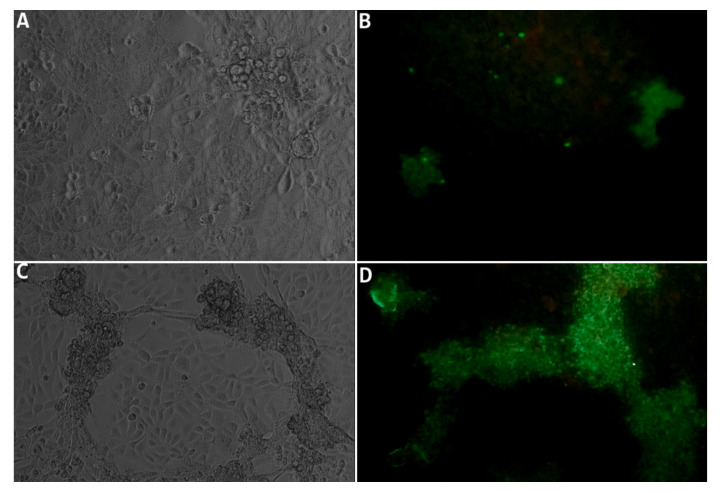
(**A**) Cytopathic effect in RK13 cells infected with an eyelid homogenate of the rabbit Male, three days after the infection, consisting of smaller aggregates of round and refringent cells, surrounded by normal cells (100×). (**B**) Positive indirect immunofluorescence staining (IFI) of recombinant MYXV infected RK13 cells three days after the inoculation (100×) (**C**) Cytopathic effect in RK13 cells infected with an eyelid homogenate of the rabbit male, six days after the infection, consisting of large aggregates of round and refringent cells forming cords over the normal cell layer (100×). (**D**) Positive immunofluorescence staining of the recombinant MYXV infected RK13 cells, six days after the inoculation (100×). Images in B and D (IFI staining) do not correspond to the same zone of the non-stained cells (A and C).

**Figure 7 viruses-12-01127-f007:**
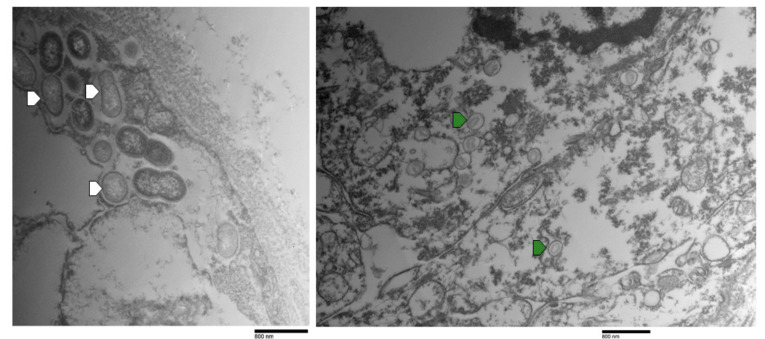
Transmission electron microscopy (TEM) micrographs. On the left, a lung cut from Female 1. At right, an eyelid epithelium cut from Female 1. White arrow heads indicate immature viral particles and green arrow heads indicate apparently mature viral particles. The black bar indicates the scale (800 nm).

**Table 1 viruses-12-01127-t001:** Summary of oligomers used for amplification and sequencing.

Primer Name	Sequence (5’-3’)	Position in MK340973 (nt)	Reference
**9B (forward)**	CGCAGGTCCACGTATAAACC	11458–11477 and 153103–153084	[15]
**9A (reverse)**	CGAACGTATCATTAGACAATG	16060–16040	
**9E (forward)**	CTTCGTCTACGCCTCCTACG	12116–12135	
**9F (reverse)**	GCGTCGTTGTGGTCAGACAGAG	15256–15243	
**305R (reverse)**	AACCCGCACAACGTAAAGTACC	12420–12399	This manuscript
**448F (forward)**	GTCATATTCCTGATTTGGGTAATC	12563–12587	
**796R (reverse)**	AGGAGGAAAAGAACCTATGACAC	12911–12889	
**1003F (forward)**	GTGTGTACCTGGTGCAGAACC	13118–13138	
**1302R (reverse)**	TGAAGATGATAATGATGATGAATATCG	13417–13391	
**1467F (forward)**	TTCATCGTTTATGGGAAAATCTATG	13582–13606	
**1819R (reverse)**	GAGGGGACAGTTATGGATGTAC	13934–13913	
**2028F (forward)**	AAGATGCGTCTGTGTAACAATCC	14143–14165	
**2325R (reverse)**	AACAATGTATACACTCATGACAGTAC	14440–14415	
**2458F (forward)**	ATGGCCATCGTAAGTTGCCATG	14573–14594	
**2847R (reverse)**	CAGAGTACTTAGATTTTCTGCTAG	14962–14939	
**2954F (forward)**	ATCCATTGTTCGTCAGTAGATCG	15069–15091	

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
