# Peer review of "Detection of Recombinant Hare Myxoma Virus in Wild Rabbits (Oryctolagus cuniculus algirus)"

_viruses, 2020, doi:10.3390/v12101127_

Round 1
Reviewer 1 Report
The authors reported a possible new recombinant MYXV strain detected from two dead European rabbits in Portugal. The surveillance of MYXV in Europe and other places where it is not endemic to is important as MYXV poses a major threat to the wild animals and farming industry. In addition, it is essential to monitor the evolution of MYXV in nature as it is heavily used as a biological weapon to control wild rabbits in Australia and New Zealand. Understanding how MYXV evolves in nature could help improve the use of such a virus-vectored biological weapons in agriculture.
The most important claim is that a recombinant MYXV, which was previously known to only infect hares, was detected in two dead animals showing signatures of myxomatosis. To establish a statement that the recombinant MYXV can cause myxomatosis in wild rabbits, Koch’s postulates could be used but were not examined in the current study. For example, it is not known if the myxomatosis was caused by rec-MYXV or other strains of MYXV that possibly co-infected the animals. In addition, the authors did not perform full-genome sequencing on the viruses, it is not known if the detected virus was the same as those found in hares, even though an insertion was found. There could be other genetic events in the virus that likely contributed to its virulence in rabbits. A whole-genome sequencing would answer many of these questions. A circumstantial evidence would be to show if there is any difference in growth kinetics between the rec-MYXV and the MXYV-Lau in cell culture.
Figure 3 and 4 should be accompanied by proper controls. A staining of the normal tissue from the dead animals for example, or from healthy lab rabbits.
Figure 6 and 7 seem redundant and unnecessary, showing amino acid sequences alone is sufficient to make the point.
It is not necessary to show figure 8 unless a comparison of growth kinetics is shown between the new virus and the MYXV-Lau strain or the rec-MYXV from hares. In addition, it is important to show an uninfected control for figure 8A as a full monolayer of RK13 overgrown for 6 days could also present some cell clusters. The green filter could be removed for figure 8A as this is not a fluorescent image. Figure 8B was claimed to be a result of immunostaining, however the method was not at all mentioned in the materials and methods section or in figure legends. At 10 days there seem to be plaques forming with clear holes in the center, it would make sense to show figure 8A at 10 days also.
Author Response
Reviewer 1
We thank the Reviewer overall suggestions and critical opinion, that were very important for us and to ultimately work better, clearer and scientifically correct.
The authors reported a possible new recombinant MYXV strain detected from two dead European rabbits in Portugal. The surveillance of MYXV in Europe and other places where it is not endemic to is important as MYXV poses a major threat to the wild animals and farming industry. In addition, it is essential to monitor the evolution of MYXV in nature as it is heavily used as a biological weapon to control wild rabbits in Australia and New Zealand. Understanding how MYXV evolves in nature could help improve the use of such a virus-vectored biological weapons in agriculture.
The most important claim is that a recombinant MYXV, which was previously known to only infect hares, was detected in two dead animals showing signatures of myxomatosis. To establish a statement that the recombinant MYXV can cause myxomatosis in wild rabbits, Koch’s postulates could be used but were not examined in the current study.
We appreciate the Reviewer 1's suggestion. In our country animal experimentation studies are quite conditioned, and approval for this case would be extremely difficult since there are other mechanisms of perceiving the pathophysiology of this disease. In addition to the clinical signs, other data was obtained namely, histopathological data, distribution of the virus in various tissues, electron microscopy data, and successful isolation of the virus in rabbit cell cultures (RK13). Therefore, we consider that there is sufficient evidence to claim that the recombinant virus was detected in these rabbits, and that they developed an infection and disease. In fact, neither electron microscopy, nor histopathology was performed in the first report of this virus in the Iberian hare, published by Viruses Journal (doi:10.3390/v11060530).
For example, it is not known if the myxomatosis was caused by rec-MYXV or other strains of MYXV that possibly co-infected the animals.
We used both a real-time PCR method directed to m000.5 L/R which is conserved in all MYXV strains. In other way, we used a conventional PCR system generates two different amplicons; a 300bp fragment for classic strains and a 3100bp fragment in recombinant MYXV. For this reason, we are really sure that just recombinant MYXV infected these rabbits.
In addition, the authors did not perform full-genome sequencing on the viruses, it is not known if the detected virus was the same as those found in hares, even though an insertion was found. There could be other genetic events in the virus that likely contributed to its virulence in rabbits. A whole-genome sequencing would answer many of these questions. A circumstantial evidence would be to show if there is any difference in growth kinetics between the rec-MYXV and the MXYV-Lau in cell culture.
Full genome sequencing is not (yet) available in our laboratory and it is impossible to request this service externally due to financial restrictions. However, we have this purpose in mind for the future. Nonetheless, we believe that this should not delay this publication as it gathers relevant data for the community. Despite the isolation of this recombinant virus in RK13 cells was carried out, it is not possible to compare data with the Lausanne strain since we do not have it. Although we could try to compare our data with data from available studies, the other variables (such as the cell line, virus strain, number of passages), could introduce noise and biased the analysis.
Figure 3 and 4 should be accompanied by proper controls. A staining of the normal tissue from the dead animals for example, or from healthy lab rabbits.
We understand the suggestion of Reviewer and we thank the willingness to improve our work. We did not include images of negative controls because the normal tissue is the same in all mammals. We consider that histopathology images are especially useful for those who have knowledge in the area, so for them, the inclusion of normal tissues is absolutely dispensable.
Figure 6 and 7 seem redundant and unnecessary, showing amino acid sequences alone is sufficient to make the point.
We understand the Reviewer 1 opinion and for this reason we allocated this two figures to supplementary data. In our opinion these alignments provide straight full information on the comparison of the nucleotide and amino acid sequences of gene M066L that is of special importance given that the gene that served as the "fitting" for insertion into the normal gene.
It is not necessary to show figure 8 unless a comparison of growth kinetics is shown between the new virus and the MYXV-Lau strain or the rec-MYXV from hares. In addition, it is important to show an uninfected control for figure 8A as a full monolayer of RK13 overgrown for 6 days could also present some cell clusters. The green filter could be removed for figure 8A as this is not a fluorescent image. Figure 8B was claimed to be a result of immunostaining, however the method was not at all mentioned in the materials and methods section or in figure legends. At 10 days there seem to be plaques forming with clear holes in the center, it would make sense to show figure 8A at 10 days also.
Currently we are unable to perform additional infection assays for a few months because INIAV BSL-2 facilities are exclusively dedicated to COVID-19 diagnosis. For this reason, we modified the figure according to Reviewer 1 concerns. The purpose of this figure is to show an image of advanced CPE caused by recombinant myxoma virus and an image of the IFI test, that to our best knowledge, is not yet available in the literature.
Reviewer 2 Report
Authors present a manuscript reporting the presence of a recombinant MYXV (rec-MYXV or MYXV-Tol) in a group of six rabbits found in the South of Portugal and showing the classical symptoms of myxomatosis. Based on comparisons with the recombinant MYXV previously found only in Iberian hares, the authors describe a characteristic 2.8 kb recombinant region of this new strain that resulted from an insertion of four novel poxviral genes towards the 5’ end of the MYXV genome. The research topic is of high veterinary importance, reporting for the first time the presence of this recombinant MYXV in rabbits, associated with high mortality.
There are, however, some issues to be addressed. In order to support the findings that the rabbits presented the recombinant MYXV previously found only in Iberian hares, the authors must present the accession numbers of the sequences obtained by PCR. Microscopy and sequence analyses of the recombinant region highlighted the presence of MYXV-Tol and further infection studies in RK13 showed infectivity. Clarification on virus detection in RK13 to rule out a cross-contamination and minor changes are suggested. Moreover, the scientific value could be significantly increased by adding at least one additional paragraph in the discussion section about vaccination programs in domestic/wild rabbits in Portugal/Spain and how this program might influence the level of susceptibility of rabbits to the new recombinant MYXV strain.
Suggested revisions:
- Page 1: Please include a reference that outlines the important predators of rabbits.
- Page 1: Line 40: Should “architecture” be “architect”?
- Page 1: Line 41: replace “has an economic and social importance as a favorite game species in Europe.” with “Is an important game species economically and socially.”
- Page 2, ln 12: please substitute “presence of an additional 2.8Kb region disrupting the M009L gene” for “presence of an additional 2.8Kb region within the M009L gene”.
- Page 2: Line 45: Please include “family” before Poxviridae
- Page 2: Line 61-65: This is a very long sentence, please split into two to make it more easily readable.
- Page 2: Line 67-71: MYXV-Tol has only been detected in a few rabbits despite extensive screening, therefore this could indicate a number of things including these positive rabbits may have already been immunocompromised and therefore MYXV-Tol may still not be optimally adapted to rabbits since it is not at present commonly identified in rabbits. Please discuss this further.
- Page 3, ln 40: Again, the referenced studies showed that the M009L gene was disrupted by an insertion of four nucleotides and not by the insertion of 2.8 kb region. Please rewrite this sentence.
- 6, line 34: Please rewrite the sentence.
- Pag 6, line 56: Since the authors showed in this manuscript that this new MYXV-Tol strain is also able to infect and cause disease in rabbits as well as hares, the term “ha-MYXV” should not be used in the manuscript.
- Page 7: Line 222: Add an “s” to “specie”
- Page 8: Line 241: Can you confirm whether PCR that was used was specific to the recombinant region to rule out cross-contamination with other MYXV strains?
- Page 9: Line 272-275: Without a comparative study looking at viral loads in infected hares this statement is not well supported. Further with only two rabbits investigated and therefore several other factors could contribute to the infection level and disease outcome.
- Page 9: Line 277: Please include a reference for this survey if possible
- Page 9: Line 305-308: Why were these data not included in your results here?
- Figures: please include the microscopic amplification.
Author Response
Reviewer 2
We thank the Reviewer overall suggestions and critical opinion, that were very important for us and to ultimately work better, clearer and scientifically correct.
Authors present a manuscript reporting the presence of a recombinant MYXV (rec-MYXV or MYXV-Tol) in a group of six rabbits found in the South of Portugal and showing the classical symptoms of myxomatosis. Based on comparisons with the recombinant MYXV previously found only in Iberian hares, the authors describe a characteristic 2.8 kb recombinant region of this new strain that resulted from an insertion of four novel poxviral genes towards the 5’ end of the MYXV genome. The research topic is of high veterinary importance, reporting for the first time the presence of this recombinant MYXV in rabbits, associated with high mortality.
There are, however, some issues to be addressed. In order to support the findings that the rabbits presented the recombinant MYXV previously found only in Iberian hares, the authors must present the accession numbers of the sequences obtained by PCR.
The sequence accession numbers are referred in lines 125 and 126: “…the two nucleotide sequences obtained were assembled using the Seqscape Software v2.7 (Applied Biosystems, Foster City, CA, USA), and submitted to GenBank (MT940239 and MT940240)."
Microscopy and sequence analyses of the recombinant region highlighted the presence of MYXV-Tol and further infection studies in RK13 showed infectivity. Clarification on virus detection in RK13 to rule out a cross-contamination and minor changes are suggested.
The DNA preparations used in the PCR and sequencing analysis were extracted from rabbit tissues. To avoid additional mutations occurring during multiplication in RK13 cells, viral isolation was used to show the presence of infectious viruses in the tissues, using as negative control mock infected cells.
Cross contamination of the rabbit materials with MYXV-positive hare tissues was prevented and ruled out by: 1) observation of histopathological lesions in the rabbits suggestive of MYXV infection; 2) qPCR results were reproduced twice from two independent extractions; 3) high viral charges obtained in the tissues by qPCR, while contamination is usually associated with high Cq values; 4) observation of MYXV particles within skin cells by transmission electron microscopy and 5) all procedures were performed under the quality policies followed in our laboratory (National Reference Laboratory for Animal Health), namely the separation between dirty and clean facilities; 6) hare analyses were not performed during the period in which this study took place.
Moreover, the scientific value could be significantly increased by adding at least one additional paragraph in the discussion section about vaccination programs in domestic/wild rabbits in Portugal/Spain and how this program might influence the level of susceptibility of rabbits to the new recombinant MYXV strain.
We thank the Reviewer 2 for this suggestion. We extended the discussion with the following paragraph: “Besides the vaccination presents good effectiveness in the industry, presenting generally a seroconversion of almost 100% [25], vaccination of wild populations is almost impossible. Other concernings Another major concern arises from the emergence of this new strain circulating in wild populations. The adaptation of wild populations to classical strains has been going on over the years [26], so the appearance of so differentiated strains theoretically poses a great risk to the angry rabbit, already threatened with extinction.”
Suggested revisions:
- Page 1: Please include a reference that outlines the important predators of rabbits.
Thank you. The reference was added.
- Page 1: Line 40: Should “architecture” be “architect”?
Thank you. Correction was done.
- Page 1: Line 41: replace “has an economic and social importance as a favorite game species in Europe.” with “Is an important game species economically and socially.”
We replaced the sentence as required.
- Page 2, ln 12: please substitute “presence of an additional 2.8Kb region disrupting the M009Lgene” for “presence of an additional 2.8Kb region within the M009Lgene”.
We thank Reviewer 2 for this suggestion.
- Page 2: Line 45: Please include “family” before Poxviridae
We included the word “family”.
- Page 2: Line 61-65: This is a very long sentence, please split into two to make it more easily readable.
We thank Reviewer 2 for point out this grammar error. The sentence was divided in two shorter sentences.
Page 2: Line 67-71: MYXV-Tol has only been detected in a few rabbits despite extensive screening, therefore this could indicate a number of things including these positive rabbits may have already been immunocompromised and therefore MYXV-Tol may still not be optimally adapted to rabbits since it is not at present commonly identified in rabbits. Please discuss this further.
We agree with the reviewer and this possibility was further addressed in lines 298-304. It can now be read: “During the two months in which the two rabbits were collected, only a small number of found dead rabbits with myxomatosis arrived at the laboratory from mainland Portugal, limiting any inference about the prevalence, frequency and distribution of the recombinant MYXV in the wild. However, since the natural recombinant MYXV emergence in 2018, and according to data collected under the +Coelho project, Beja was the district from which more hares were sampled (52 out of 170) and tested, and was also one of the districts mostly affected by myxomatosis (34.6% of positivity in the sample).”
- Page 3, ln 40: Again, the referenced studies showed that the M009Lgene was disrupted by an insertion of four nucleotides and not by the insertion of 2.8 kb region. Please rewrite this sentence.
We could not understand the reviewer's comment. In fact, an insertion of 2.8kb within the M009 gene, and this is described in all references.
- 6, line 34: Please rewrite the sentence.
We thank the Reviewer.
- Pag 6, line 56: Since the authors showed in this manuscript that this new MYXV-Tol strain is also able to infect and cause disease in rabbits as well as hares, the term “ha-MYXV” should not be used in the manuscript.
We understand the Reviewer concerns. In fact, in the Introduction we point this issue “The detection of a recombinant MYXV in hares, and the apparent segregated circulation of classical MYXV in rabbits and ha-MYXV in hares, initially suggested the adaptation of MYXV to hares in order to efficiently multiply in this species. Given that hare MYXV, originally considered hare specific, is also being detected in rabbits, a more generalist designation, geographic and species independent, such as rec-MYXV (for recombinant myxoma virus) may be preferable for the future.”
However, until new evidences (by complete genome sequencing) are gathered, we decided to adder to the nomenclature to avoid confusion.
- Page 7: Line 222: Add an “s” to “specie”
Thank you. We corrected this mistake.
- Page 8: Line 241: Can you confirm whether PCR that was used was specific to the recombinant region to rule out cross-contamination with other MYXV strains?
We corrected “qPCR” to “conventional PCR”. We confirm that his system generates two different amplicons; a 300bp fragment for classic strains and a 3100bp fragment in recombinant MYXV.
- Page 9: Line 272-275: Without a comparative study looking at viral loads in infected hares this statement is not well supported. Further with only two rabbits investigated and therefore several other factors could contribute to the infection level and disease outcome.
We understand the Reviewer concerns. During the motioned national survey, we analyzed several dozens of hares and rabbits with myxomatosis. In both rabbits, myxomatosis frequently leads to poor body status and anatomopathological lesions with more chronic characteristics. Because this has not occurred in these two cases, we discussed this as a possibility. Nonetheless, the emergence of new strains in rabbits generally induces more several disease in the first times, it was happened for example with MYXV in Australia.
- Page 9: Line 277: Please include a reference for this survey if possible
Thank you. A reference was included.
- Page 9: Line 305-308: Why were these data not included in your results here?
These results were obtained days before the submission and we did not want to delay it. We included the available information now.
- Figures: please include the microscopic amplification.
The histopathologic figures have amplification 40x, or 100x that corresponds to total magnification. This information was added to the figures.